# Cloning and Characterization of Chitin Deacetylase from *Euphausia superba*

**DOI:** 10.3390/ijms25042075

**Published:** 2024-02-08

**Authors:** Xutong Wang, Jiahao Tan, Huaying Zou, Fang Wang, Jiakun Xu

**Affiliations:** 1College of Food Science and Technology, Shanghai Ocean University, Shanghai 201306, China; 2State Key Laboratory of Mariculture Biobreeding and Sustainable Goods, Yellow Sea Fisheries Research Institute, Chinese Academy of Fishery Sciences, Laboratory for Marine Drugs and Byproducts of Pilot National Lab for Marine Science and Technology, Qingdao 266071, China; 3Key Laboratory of Sustainable Development of Polar Fisheries, Ministry of Agriculture and Rural Affairs, Qingdao 266071, China

**Keywords:** chitin deacetylase, *Euphausia superba*, heterologous expression, enzymatic properties, molecular simulation

## Abstract

Chitin deacetylase (CDA) can catalyze the deacetylation of chitin to produce chitosan. In this study, we identified and characterized a chitin deacetylase gene from *Euphausia superba* (*EsCDA-9k*), and a soluble recombinant protein chitin deacetylase from *Euphausia superba* of molecular weight 45 kDa was cloned, expressed, and purified. The full-length cDNA sequence of *Es*CDA-9k was 1068 bp long and encoded 355 amino acid residues that contained the typical domain structure of carbohydrate esterase family 4. The predicted three-dimensional structure of *Es*CDA-9k showed a 67.32% homology with *Penaeus monodon*. Recombinant chitin deacetylase had the highest activity at 40 °C and pH 8.0 in Tris-HCl buffer. The enzyme activity was enhanced by metal ions Co^2+^, Fe^3+^, Ca^2+^, and Na^+^, while it was inhibited by Zn^2+^, Ba^2+^, Mg^2+^, and EDTA. Molecular simulation of *Es*CDA-9k was conducted based on sequence alignment and homology modeling. The *Es*CDA-9k F18G mutant showed a 1.6-fold higher activity than the wild-type enzyme. In summary, this is the first report of the cloning and heterologous expression of the chitin deacetylase gene in *Euphausia superba*. The characterization and function study of *Es*CDA-9k will serve as an important reference point for future application.

## 1. Introduction

Chitin is the second most abundant natural polymer after cellulose and the most abundant renewable nitrogen-containing substance on Earth [1,2]. Chitin is widely present in the cell walls of crustaceans, insect exoskeletons, mollusks, and fungi [3,4]. Natural chitin has a dense crystal structure, highly ordered three-dimensional network, and high degree of polymerization and insolubility in conventional solvents, which seriously limits its practical application [5,6]. Chitosan is a deacetylation product of chitin; due to the presence of amino groups, chitosan has better solubility than chitin [7,8]. The traditional industrial preparation method of chitosan is to treat chitin under alkaline conditions; it could cause serious environmental pollution and is difficult to control the reaction process [9,10]. The homogeneity of produced chitosan is poor, and it cannot meet the requirements of biopharmaceuticals for high-quality chitosan [1,11,12].

Chitin deacetylase (CDA) belongs to the carbohydrate esterase 4 (CE4) superfamily [6]. It can catalyze the deacetylation of chitin, thereby preparing chitosan with better biocompatibility and biodegradability [11,13]. Therefore, the enzymatic deacetylation of chitin using chitin deacetylases offers an environmentally friendly alternative for chitosan production (Figure 1) [9,14,15,16,17]. In the past, most studies on chitin deacetylase have focused on strain screening, enzymatic properties, and purification, which were generally obtained from bacteria, fungi, and marine microorganisms. CDA was first discovered from extracts of fungus *Mucor rouxii* [18], and then several different chitin deacetylases have been identified, purified, and characterized from other fungi such as *Absidia coerulea* [19], *Colletotrichum lindemuthianum* [20], *Saccharomyces cerevisiae Cda2p* [21], *Scopulariopsis brevicaulis* [22], *Rhizpus circinans* [23], *Mortierella* sp. DY-52 [24], and *Flammulina velutipes* [25]. Until now, there have been few reports of chitin deacetylase obtained from crustacean, and only a few CDA genes have been determined in crustaceans, including *Pm*CDA1 from *Penaeus monodon* [26], *Pc*CDA1 from *Palaemon carinicauda* [27], and *Es*CDA-l from *Eriocheir sinensis* [28]. Therefore, little is known about the characteristics and underlying molecular mechanisms of crustacean CDA.

*Euphausia superba* is a small marine planktonic crustacean in the Antarctic waters of the Southern Ocean [29]. It is the most abundant animal on the planet, with a biomass of 300–500 million tons, and is central to the Antarctic food web [30,31]. The *Euphausia superba* fishery is the largest fishery in the Southern Ocean, and it has been operating for over 47 years [32,33]. It is one of the few pelagic crustacean that can be fished commercially [34]. *Euphausia superba* has a variety of active substances, such as lipids, proteins, and enzymes [35]. Here, we present the cloning, heterologous expression, purification, and characterization of chitin deacetylase from *Euphausia superba*, and investigate the optimum temperature, pH, and stability of *Es*CDA-9k. The molecular simulation and site-directed mutagenesis were also conducted for better catalytic ability. This study provides an experimental basis for the molecular mechanism of the deacetylation of chitin. To our knowledge, this is the first report on the heterologous expression of CDA from *Euphausia superba*.

## 2. Results

### 2.1. Cloning and Bioinformatics Analysis

The krill cDNA was used as a template to verify the *EsCDA-9k* gene by subcloning. The total length of *Es*CDA-9k cDNA was 1068 bp, it contained 355 AA, and its molecular weight (Mw) was 40,864.96 Da. The *Es*CDA-9k theoretical pI was 5.26 and included 40 positively charged residues (Arg+Lys) and 25 negatively charged residues (Asp+Glu). The instability index was 47.54 (>40), indicating that the *Es*CDA-9k protein was unstable. The aliphatic index was 60.39, indicating that the protein was soluble in lipids. The sequences obtained by NCBI Blast were used to obtain some sequences that were highly similar to those of *Es*CDA-9k. The highest similarity was found in CDAs from *Penaeus monodon* (67.32%), followed by *Penaeus vannamei* (66.73%), *White shrimp* (66.48%), *Chinese shrimp* (65.92%), and *Drosophila melanogaster* (64.99%). SOMPMA was used to predict the two-dimensional structure of proteins, and its secondary structure mainly included 24.93% α-helix, 54.85% random coil, 14.68% extended strand, and 5.54% β-turn. SignalP-5.0 analyzed the signal peptide of the protein, with a probability of 0.0007 for the signal peptide (Sec/SPI), indicating that the protein has no signal peptide (Appendix A). The *Es*CDA-9k has typical characteristics of CE4 family enzymes. NetPhos 3.1 was used to analyze the phosphorylation sites of proteins, and it was found that the phosphorylation of proteins was mainly concentrated in the Tyr, Ser, and Thr residues of the peptide chain, with a total of 21 phosphorylation sites (Appendix A). The glycosylation sites of the protein were analyzed by NetNGlyc-1.0 and NetOGlyc-4.0, and it was found that the protein contained four N-glycosylation sites (36NQTL, 59NYSA, 81NSST, 111NTTD) and one O-glycosylation site (Appendix A). The amino acid is a mature protein sequence; it has no signal peptide and has phosphorylation and glycosylation modifications. The amino acid sequences encoded by *Es*CDA-9k were compared to the multi-sequence homology of each species using DNAMAN8.0 (Figure 2) and the phylogenetic tree was constructed using MEGA (Appendix A). The phylogenetic tree showed that deacetylases from three different clades: the CE4 superfamily, uraD_N-term-dom superfamily, and spore_pdaA superfamily, and *Es*CDA-9k belong to the CE4 superfamily. The CE4 superfamily members generally include the NodB homologous domain and the metal-binding triad (His-His-Asp) structure. The uraD_N-term-dom superfamily exhibits significant differences in oligomer binding and active site geometry. For example, the metal-binding triad (His-His-Asp) replaced Glu-His-Trp in the uraD_N-term-dom superfamily. The spore_pdaA superfamily contains an α/β-related protein with unique Lys and Arg residues in protein concavity compared to the CE4 superfamily. 

### 2.2. Protein Expression and Purification

In order to improve protein translation efficiency and thereby enhance protein expression, we adjusted the synonymous codons of genes according to the codon preferences of the host. Therefore, the gene encoding the mature protein was optimized based on the preference of the host codon. The recombinant plasmid was constructed and expressed in *P. pastoris* GS115, as described in Section 4.3. After methanol induction for 5 days, the crude enzyme was loaded onto an Ni-NTA superflow device for purification. The results showed that the protein solution corresponding to the peak No. III had enzyme activity (Appendix A). SDS-PAGE showed a single band of purified protein with a relative molecular mass of 45 kDa (Figure 3). The molecular weight of protein bands was slightly higher than the theoretical value, due to the presence of glycosylation sites and His-tags in the enzyme. The His-tags have a low molecular weight and do not affect protein structure and functions. This means that it is not necessary to separate the His-tag from the target protein. *Es*CDA-9k was purified from 100 mL crude extracellular enzyme solution using HisTrap HP purification on the Ni-NTA superflow (Table 1). That is, the *Es*CDA-9k protein was purified 1.3-fold by HisTrap HP affinity chromatography. The specific activity of the purified enzyme was 24.2 U/mg.

### 2.3. Effects of Temperature and pH on the Activity and Stability of EsCDA-9k

The influence of temperature on enzyme activity was determined in the temperature range of 20–60 °C (Figure 4a). The optimal temperature of *Es*CDA-9k was 40 °C. As shown in Figure 4b, 88.5%, 84.1%, 62.4%, 38.4%, and 35.2% of initial activity was retained when the enzyme solution was maintained at 4, 25, 35, 40, and 45 °C for 10 h. *Es*CDA-9k has a relatively lower optical temperature compared to other sources. For example, the optimum temperatures of CDAs from *Aspergillus nidulans* [36], *Scopulariopsis brevicaulis* [22], and *Colletotrichum lindemuthianum* [20] were 50, 55, and 60 °C, respectively. pH is a major factor affecting the activity of enzymes. The acidity and alkalinity of the buffer may cause changes in the conformation of the enzyme, and the degree of dissociation of basic functional groups in enzyme activity centers, thereby affecting enzyme activity [37]. The effect of pH on *Es*CDA-9k enzyme activity was determined (Figure 4c), and the results showed that the optimal pH of *Es*CDA-9k was 8.0. The types of solution can also have a significant impact on the catalytic ability, and proteins exhibit relatively good catalytic activity in Tris-HCl solution. In the pH stability test (Figure 4d), the enzyme activities could maintain 91.2%, 85.3%, and 83.2% of their original activity after 10 h of incubation during pH 7.0–9.0. As acidity increased, the enzyme activity gradually decreased. At pH 5, the enzyme activity remained 38.4% after 10 h of incubation. The optimum pH of CDA from a different source was quite similar, and the optimum pH in most of the previous reports on chitin deacetylases ranged from pH 7.0 to 9.0. For example, the optimum pH of CDAs from *Flammulina velutipes* [25], *Saccharomyces cerevisiae Cda2p* [21], and *Metarhizium anisopliae* [38] was 7.0, 8.0, and 8.5, respectively. Like most extracellular enzymes, *Es*CDA-9k was an alkaline chitin deacetylase with good pH stability.

### 2.4. Effects of Chemical Reagents on EsCDA-9k Enzyme Activity

In order to test the stability of the chitin deacetylase *Es*CDA-9k against metal ions, metal ions (K^+^, Na^+^, Ba^2+^, Ca^2+^, Cu^2+^, Mg^2+^, Zn^2+^, Co^2+^, and Fe^3+^), and EDTA were added at 1 mM or 10 mM, respectively, into the reaction system. The relative enzyme activity after the influence of metal ions, such as Co^2+^, Fe^3+^, Ca^2+^, and Na^+^, was 118.3%, 110.1%, 107.2%, and 108.3% (Figure 5). Zn^2+^, Ba^2+^, and Mg^2+^ inhibited the activity of *Es*CDA-9k, with a relative enzyme activity of 68.2%, 85.6%, and 84.8%, respectively; K^+^ and Cu^2+^ had minimal effects on the enzyme activity, with a relative enzyme activity of 97.6% and 98.3%. Furthermore, the addition of EDTA inhibited the *Es*CDA-9k activity, with a relative enzyme activity of 53.6%. In previous reports, it was also found that the enzyme activity of chitin deacetylase could be enhanced in the presence of Ca^2+^ and Co^2+^ [39,40].

### 2.5. Site-Directed Mutagenesis and Comparison of Enzyme Activity of Mutants

The homology model (Figure 6a) consists of a highly conserved (α/β)_8_ folded barrel structure and six loops. The 3D protein structure (Figure 6b) shows that key residues that may contribute to catalytic function, including the metal-binding triad (His-His-Asp) around Zn^2+^, and Asp19 is the general acid residue for catalysis [41]. Phe124 and Gln246 are around the catalytic site of the metal-binding triad, and Phe18 and Arg121 are located in a conservative area. Furthermore, Phe18 and Phe124 are located near the protein ligand pocket. Based on the molecular docking results, as well as the size, polarity, and side-chain properties of the amino acids, we mutated Phe to the smaller Gly (Figure 7) in order to allow better substrate passage through the channel. We mutated basic amino acid Arg121 and aliphatic amino acid Gln246 to acidic amino acid Asp121 and aromatic amino acid His246 (Figure 8), respectively. Therefore, we selected F18G, R121D, F124G, and Q246H for targeted mutation to search for mutants with higher enzyme activity. All four prepared single mutants showed improved catalytic performance (Table 2), with the F18G mutant showing a relative enzyme activity of 160%, the F124G mutant showing a relative enzyme activity of 141%, the R121D mutant showing a relative enzyme activity of 124%, and the Q246H mutant showing a relative enzyme activity of 135%, respectively.

### 2.6. Scanning Electron Microscope Analysis

As shown in Figure 9, the α-chitin showed a prominent arranged microfibrillar crystalline structure and a smooth surface in SEM (Figure 9a), which was absent in the *Es*CDA-9k-treated α-chitin. The *Es*CDA-9k-treated α-chitin surface microstructure was damaged, became non-smooth, and exhibited grooves (Figure 9b). The above results indicate that *Es*CDA-9k has a good deacetylation effect.

### 2.7. Fourier Transform-Infrared (FT-IR) Spectroscopy

The results of infrared spectroscopy of α-chitin and *Es*CDA-9k-treated α-chitin are shown in Figure 10. There are some characteristic peaks in both spectra: the peak at 3400 cm^−1^ could be attributed to the -NH_2_ and -OH groups stretching vibration and intermolecular hydrogen bonding [42]. In α-chitin spectra (Figure 10b), the band at 1655 cm^−1^ corresponds to the stretching of amide I, and the absorption bands at 1550 and 1380 cm^−1^ correspond to amide II (N-H bending) and amide III (C-N stretching), respectively [43]. After the deacetylation of chitin (Figure 10a), the multiple absorption peak broadening at 3450 cm^−1^ is due to the overlap of the -OH stretching vibration peak and the -NH stretching vibration peak. It is usually believed that the ratio of the absorption peak at 1655 cm^−1^ to the absorption peak at 3450 cm^−1^ (A_1655_/A_3450_) is linear with the degree of deacetylation of chitin, so the ratio of sample deacetylation can be calculated using this ratio [44]. The degree of deacetylation (DD) of α-chitin and *Es*CDA-9k-treated α-chitin was observed to be 27% and 52%, respectively. The results indicated that chitin deacetylation was enhanced through this enzymatic pretreatment.

## 3. Discussion

In this study, we achieved for the first time the efficient secretory expression of *Es*CDA-9k in the eukaryotic expression system of *Pichia pastoris*. The *Pichia pastoris* expression system involved in this study has the advantages of efficient expression of active proteins, easy purification of target proteins, and is suitable for high-density fermentation. It simplifies the separation and purification process of the recombinant enzyme, and saves the cost of industrial application of the recombinant enzyme.

*Es*CDA-9k exhibits the highest activity at 40 °C and pH 8.0 in Tris-HCl buffer. It exhibits enzyme activity between 20 and 60 °C, and the enzyme activity changes show a trend of increasing and then decreasing, and reached a maximum at 40 °C (Figure 4a). At present, the optimal catalytic temperature for chitin deacetylases from other sources is mostly 50–70 °C. *Es*CDA-9k has low-temperature catalytic activity and stability, which can save heating energy in industrial applications and is beneficial for practical applications. The enzyme activity increased with the increase in pH between pH 3.0 and 8.0, while the enzyme activity gradually decreased at a pH higher than 8.0 (Figure 4c). Therefore, this enzyme is a weakly alkaline enzyme with high activity under neutral and weakly alkaline conditions. In addition, the enzyme activity decreased significantly in glycine-NaOH buffer, perhaps due to the acid-base neutralization reaction between the hydroxide ions in the solution and the acetic acid in the product. The enzyme activity was enhanced by metal ions Co^2+^, Fe^3+^, Ca^2+^, and Na^+^, while it was inhibited by Zn^2+^, Ba^2+^, Mg^2+^, and EDTA. EDTA as a typical metalloenzymes inhibitor, could form stable complexes with metal ions, leading to a decrease in enzyme activity. Currently, most of the CDAs discovered are glycoproteins with certain similarities; there are some differences in the properties of CDAs from different sources, such as their optimal temperature, optimal pH, molecular weight (Appendix A). The optimal temperature and pH of chitin deacetylase from *Mucor rouxii* are 50 °C and 7.0, respectively [18]. The optimal temperature and pH of CDA found in *Penicillium oxalicum* SAEM-51 are 50 °C and 9.0, respectively; Co^2+^ and Ca^2+^ have a promoting effect, while Mn^2+^ has an inhibitory effect [40]. The optimal temperature and pH of CDA from *Saccharomyces cerevisiae* are 50 °C and 7.0, respectively, and Co^2+^ has a promoting effect [45]. The optimal temperature and pH of CDA1 from *Coprinopsis cinerea* are 70 °C and 7.0, respectively, and Zn^2+^, Co^2+^, and Cu^2+^ have a promoting effect [46]. The K_m_ and *k*_cat_ of *Es*CDA-9k are 0.54 mM and 1.72 S^−1^, respectively. The catalytic ability of *Es*CDA-9k is higher than the *An*CDA from *Aspergillus nidulans* (K_m_ = 0.072 mM, *k*_cat_ = 1.4 S^−1^) [47], and the *Vc*CDA from *Vibrio cholerae* (K_m_ = 0.40 mM, *k*_cat_ = 0.58 S^−1^) [48].

We conducted docking simulation to investigate the characteristics of the interaction between *Es*CDA-9k and chitin molecules. The mutation of phenylalanine in F18G and F124G mutants to smaller glycine reduced the spatial bit resistance and the enzyme binding domain space became larger, which could have facilitated the access of the substrate into the active center of the protein (Figure 7). The distance of the R121D and Q246H mutant amino acids from the substrate was shortened compared to the wild-type; R121D decreased from 5.1 to 4.5 Å (Figure 8a), and Q246H decreased from 4.3 to 3.9 Å (Figure 8b). The mutation of arginine at the 121 position to aspartic acid in the R121D mutant resulted in shorter side chains, which enlarged the cavity of the binding domains. The distance from histidine 246 to the substrate became shorter in the Q246H mutant, probably favoring substrate binding to the enzyme. Therefore, the arrangement of amino acid residues in the active site has an obvious influence on the catalytic activity of the enzyme.

## 4. Materials and Methods

### 4.1. Materials

All chemicals were purchased from Sinopharm Reagent Co., Ltd. (Shanghai, China), Shanghai Sangon Biology Co., Ltd. (Shanghai, China), and Dalian Takara Co., Ltd. (Dalian, China). Other chemical reagents used in this study were of analytical grade unless specifically indicated. Mettler Toledo S20 pH meter (Mettler Toledo, Greifensee, Switzerland), HH-1 electric constant temperature water bath (Changzhou Guohua, Changzhou, China), CR21GII high-speed refrigerated centrifuge (Hitachi, Tokyo, Japan), AKTA Explorer FPLC System (GE, Milwaukee, WI, USA), Ni-NTA superflow (Duesseldorf, Germany), gelelectrophoresis system (Bio-Rad, Hercules, CA, USA), and ultrapure water purification system (Millipore, Bedford, MA, USA) were used.

### 4.2. Construction of Recombinant Plasmid

RNA was isolated from whole animals using the RNeasy system with column DNase treatment (Qiagen, Beijing, China). Genomic DNA was removed from total RNA by treatment with RNA-Free DNase (Thermo, Waltham, MA, USA). The *Euphausia superba* RNA, as a template, was sent to BGI Company for transcriptome sequencing, and the gene sequence of *Euphausia superba* chitin deacetylase was spliced from the sequencing results by gene mining and NCBI library comparison. Finally, the complete sequence of the *Es*CDA-9k gene was obtained. Reverse transcription reactions were performed using the PrimeScript 1st Strand cDNA Synthesis Kit (TaKaRa, Kyoto, Japan). Then, the cDNA was used as a template to amplify the *Es*CDA-9k gene, which was obtained through transcriptome sequencing. The *Es*CDA-9k gene sequence was submitted to NCBI, and the number is GenBank PP061640. Two primers, *EsCDA-9k*F1 (5′-GAATTCATGCCAAACGGTATG-3′) and *EsCDA-9k*R1 (5′-CCGCTCGAGTTACGGGTTG-3′) (Appendix A) were used. The amplified product was purified and digested with EcoR I/Not I and then ligated into a EcoR I/Not I digested pPIC9K vector (Appendix A). 6×His-tag was introduced at the N-terminal end of *Es*CDA-P9; it facilitates the isolation and purification of recombinant proteins.

### 4.3. Protein Expression and Purification

The Plasmid Extraction Kit was used to recover the plasmids. After gene sequencing, correct plasmids were digested by Sal I before being transformed into *P. pastoris* GS115 competent cells. Transformant was cultured in BMGY at 30 °C with shaking at 250 rpm until OD_600_ reached 5.0. Next, the strains were collected by centrifugation at 5000× *g* for 10 min. *P. pastoris* cells were suspended in BMMY to reach OD 2.0 at 600 nm. The fermentation was initiated by supplementing 1% methanol in the medium every day and the cells were incubated at 28 °C for 5 days [49].

The purification was performed on the AKTA fast liquid chromatography system. The crude enzyme was loaded onto Ni-NTA superflow, and using phosphate loading buffer and phosphate elution buffer containing 500 mM imidazole at a flow rate of 1.0 mL/min. Ni^2+^ can bind not only to recombinant proteins with His-tags, but also to imidazole. Therefore, target proteins can be obtained that use different concentrations of imidazole. The obtained samples were ultra-filtered at 10 kD, 4000 rpm, and imidazole was removed by the successive addition of phosphate buffer. The samples were analyzed for purity using 12% SDS-PAGE.

### 4.4. Activity Assay

The enzymatic activity assay of *Es*CDA-9k was performed by measuring the content of acetic acid produced using the K-ACET acetic acid determination kit (Megazyme, Bray, Ireland). The reaction mixtures of 50 µL (GlcNAc)_2_ (10 mg/mL) and 50 µL of *Es*CDA (5.7 mg/mL) were added to 100 µL Tris-HCl buffer solution (10 mM, pH = 8.0), and incubated at 40 °C for 2 h. The enzyme reaction was terminated by boiling at 100 °C for 10 min in a sealed tube prior. The supernatant was collected by centrifugation at 12,000 rpm for 5 min, and the amount of acetic acid in 10 µL supernatant was determined to calculate the enzyme activity through the A_340_ according to the assay procedure of the Acetic Acid Assay Kit [46]. For determination of K_m_ and *k*_cat_, (GlcNAc)_2_ was used as a substrate at various concentrations (2, 4, 6, 8, 10 mM). One unit of *Es*CDA-9k activity was defined as the amount of enzyme required to catalyze the release of 1 µmol acetic acid per minute.

### 4.5. Biochemical Characterization

To determine the optimal temperature for enzymes, the enzyme activities of *Es*CDA-9k were measured under different temperature conditions, which were set at 20, 25, 30, 35, 40, 45, 50, 55, and 60 °C, respectively. The reaction system was placed in Tris-HCl buffer (10 mM, pH 7) at various temperatures, incubated for 2 h, and the highest enzyme activity was set to 100%. The enzyme activity was measured at 4, 25, 35, 40, and 45 °C after being incubated at 2, 4, 6, 8, and 10 h to determine the temperature stability of *Es*CDA-9k. Three replicate experiments were set for each group.

The effect of pH on enzyme activity was analyzed from the variation of pH, which was configured citrate buffer (10 mM, pH 3.0–5.0), phosphate buffer (10 mM, pH 5.0–7.0), Tris-HCl buffer (10 mM, pH 7.0–9.0), and glycine-NaOH buffer (10 mM, pH 9.0–10.0). The reaction system was placed in various buffers at 40 °C, incubated for 2 h, and the enzyme activity was measured. The highest enzyme activity was set to 100%, and the rest of the enzyme activity was calculated to determine the optimum pH of the enzyme. The enzyme solution and the substrate were added to each buffer in the same ratio, a water bath measured the reaction at 40 °C for 10 h, and the enzyme activity was measured to determine the pH stability. Three replicate experiments were set for each group.

Purified *Es*CDA-9k was added at various metal ions (K^+^, Na^+^, Ba^2+^, Ca^2+^, Cu^2+^, Mg^2+^, Zn^2+^, Co^2+^, and Fe^3+^; ions solution for 1 mM and 10 mM), and EDTA for analysis of the effect of metal ions on the enzyme activity of *Es*CDA-9k. The system was placed in Tris-HCl buffer (10 mM, pH 8) at 40 °C, incubated for 1 h, and the enzyme activity was measured. An equal amount of deionized water was added instead of metal ion solution, with enzyme activity set to 100%. Three replicate experiments were set for all groups.

### 4.6. Homology Modeling and Molecular Docking

The sequences of *Es*CDA-9k were submitted to SWISS-MODEL to construct its 3D structure using 5ZNT as the template [50]. The geometric conformation of the protein model was observed using PyMOL [51]. The best model was further evaluated using the PROCHECK and VERIFY3D in order to analyze the quality and consistency of the generated model. The molecular structure of (GlcNAc)_2_ was obtained from the ZINC database. The AutoDock 4.2 software was used for molecular docking. Molecular docking calculations were performed by using Autodock 4.2 with the aid of AutoDock Tools 1.5.6 [52]. Ligand position in *Es*CDA-9k protein was set at grid coordinates x = 20.6, y = 24.3, and z = 36.5. Genetic algorithm parameters were assigned at 100 GA run and population size of 150. Finally, the docking site conforming to the catalytic conditions was selected as the initial conformation for analysis.

### 4.7. Site-Directed Mutagenesis

A DNAMAN was used for amino acid sequence alignment and conservative domain analysis [53]. By sequence alignment, the amino acid residues located on the surface of the substrate-binding region excluding conserved catalytic motifs were selected; we selected 4 amino acids in the conserved region for mutation. Then, we used the Q5 site-directed mutagenesis kit for mutagenesis (Appendix A). Primers for each site are listed in Appendix A. All mutations were made by PCR-based site-directed mutagenesis and verified by DNA sequencing. After PCR and KLD reaction, the mutant plasmids were transformed into DH5α chemically competent cells for propagation of the plasmids. The Plasmid Extraction Kit was used to recover the plasmids. Then, the mutant plasmids were transformed into *P. pastoris* GS115 chemically competent cells to obtain mutant strains.

### 4.8. Scanning Electron Microscope Characterization

The α-chitin and *Es*CDA-9k-treated α-chitin were characterized as microstructure changes by scanning electron microscopy with magnification ×2000 times. The final concentration was 10 mg/mL α-chitin and enzyme solution with 10 mM Tris-HCl buffer with shaking fermentation at 40 °C for 10 h. All samples were dried to a constant weight at 100 °C and were, respectively, adhered to the metal sample stage with a conductive adhesive. The metal film was sprayed in a vacuum evaporator and observed under a scanning electron microscope at 3 kV [54].

### 4.9. Determination of the Degree of Deacetylation

The degree of deacetylation of α-chitin and *Es*CDA-9k-treated α-chitin was determined by infrared spectroscopy [55]. A dried sample of 1 mg and 100 mg of dried potassium bromide were placed to grind in a mortar. After thorough mixing, the mixture was compressed into transparent flake by a tableting machine, and analyzed by an infrared spectrometer (Nicolet iG50) in a wavenumber region of 400 to 4000 cm^−1^, with spectral resolution 0.5 cm^−1^ and 64 scans. Infrared determination of the degree of deacetylation calculation formula:$$\mathrm{DD}=(1-\frac{A_{1655}/A_{3450}}{1.33})\times 100$$

## 5. Conclusions

In this study, the CDA gene from *E. superba* was cloned. The total length of *Es*CDA-9k is 1068 bp, encoding a mature protein of 355 amino acids. We expressed the *Es*CDA-9k in *P. pastoris* GS115 and obtained pure protein. The successful expression of *Es*CDA-9k was determined through protein electrophoresis and enzyme activity assay. The optimum temperature and pH of the recombinant *Es*CDA-9k were 40 °C and 8.0, respectively. It has good stability under neutral and weakly alkaline conditions, as well as low-temperature conditions. The metal ions Co^2+^, Fe^3+^, Ca^2+^, and Na^+^ have an activation effect on *Es*CDA-9k. Furthermore, site-directed mutagenesis allowed us to verify important sites of *Es*CDA-9k activity, with mutants showing a raised catalytic activity compared to the wild-type. Our study laid an experimental foundation for the rational design, directed evolution, and application of chitin deacetylase.

## Figures and Tables

**Figure 1 ijms-25-02075-f001:**
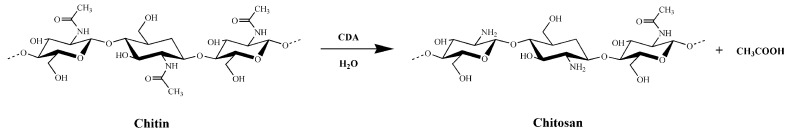
Chemical catalyzed by chitin deacetylase (CDA). CDA catalyzes the cleavage of chitin to chitosan and acetic acid.

**Figure 2 ijms-25-02075-f002:**
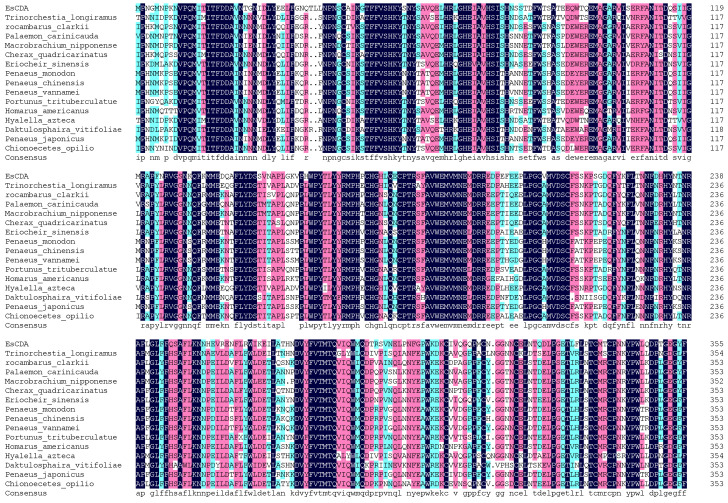
Multiple-sequence alignment of *Es*CDA-9k and other reported chitin deacetylase. Completely conservative amino acids are highlighted in black. Amino acids with a similarity of 80% and 60% are highlighted in pink and blue, respectively.

**Figure 3 ijms-25-02075-f003:**
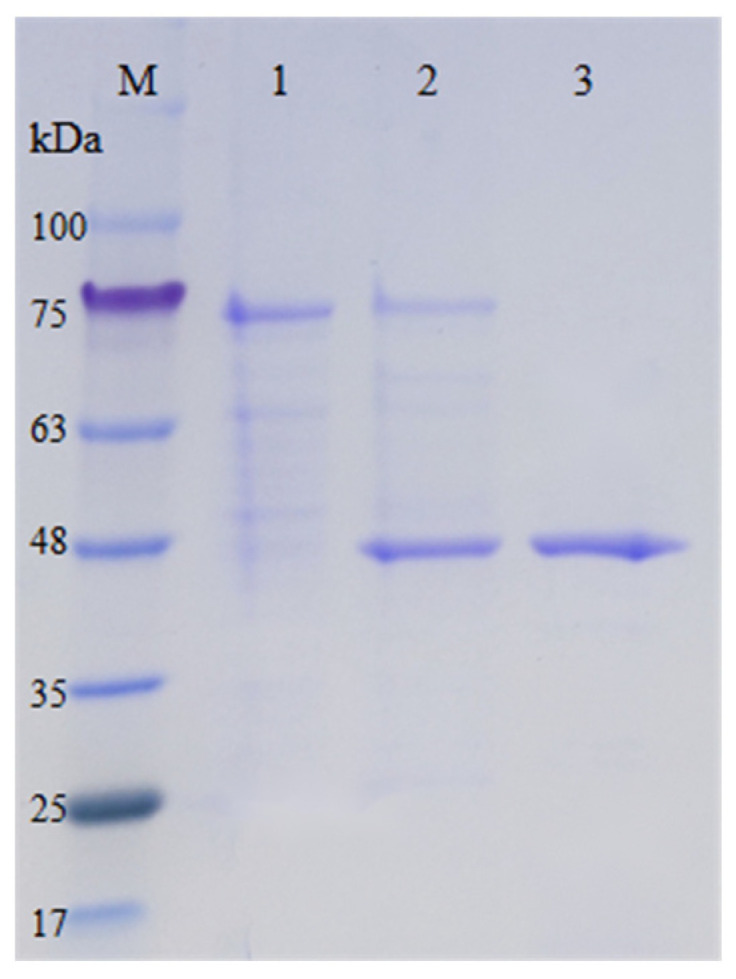
SDS-PAGE analysis of fermented broth and purified *Es*CDA-9k. Lane M: protein molecular weight markers; lane 1: supernatant of *P. pastoris* (empty vector) after 5 days induction; lane 2: supernatant of *P. pastoris* (*Es*CDA-9k) after 5 days induction; lane 3: *Es*CDA-9k purified by HisTrap HP column.

**Figure 4 ijms-25-02075-f004:**
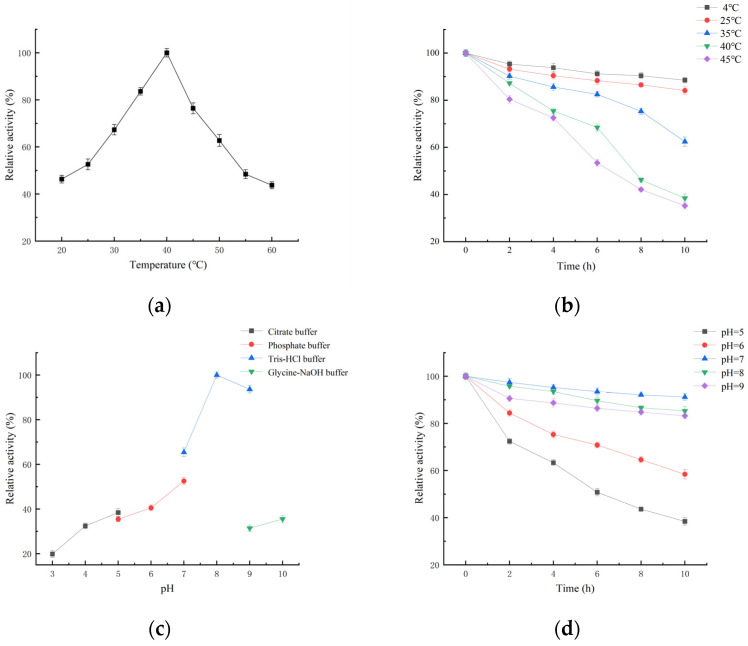
Biochemical characteristics of *Es*CDA-9k. (**a**) Effects of temperature on the activity of *Es*CDA-9k. (**b**) Effects of temperature on the stability of *Es*CDA-9k. (**c**) Effects of pH on the activity of *Es*CDA-9k. (**d**) Effects of pH on the stability of *Es*CDA-9k.

**Figure 5 ijms-25-02075-f005:**
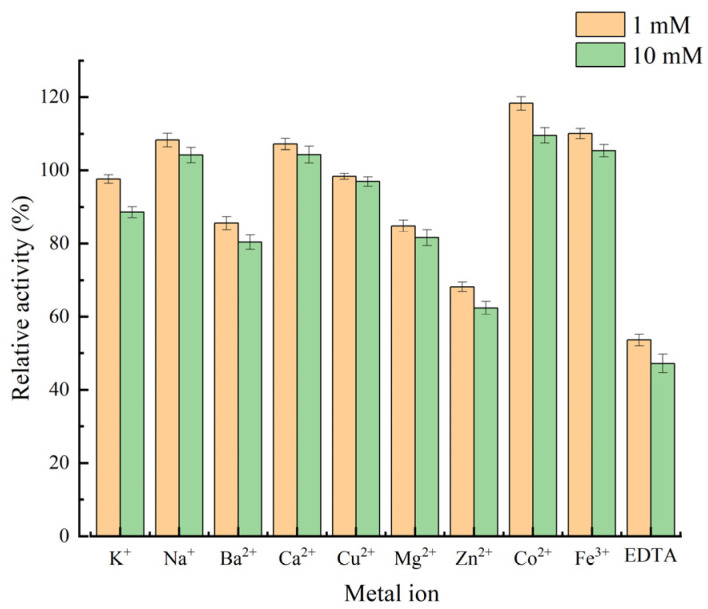
Effects of various chemical agents on enzyme activity of *Es*CDA-9k.

**Figure 6 ijms-25-02075-f006:**
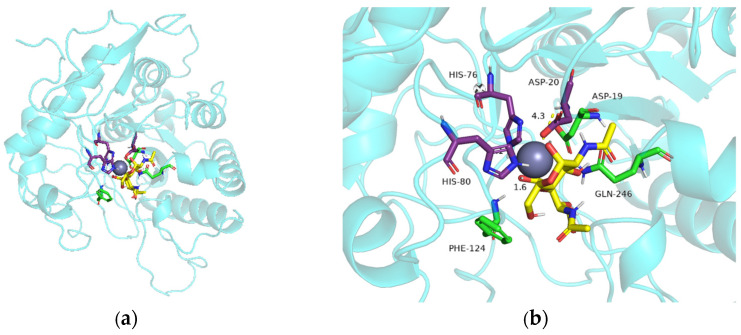
The docking simulation to the active site of (**a**) *Es*CDA-9k and (**b**) partial enlarged image. The metal-binding His-His-Asp triad (purple) and (GlcNAc)_2_ (yellow) are represented as stick models. The grey balls indicate metal ions and the amino acid residues at 19, 124, and 246 positions are shown in green. The cartoon mode shown in blue.

**Figure 7 ijms-25-02075-f007:**
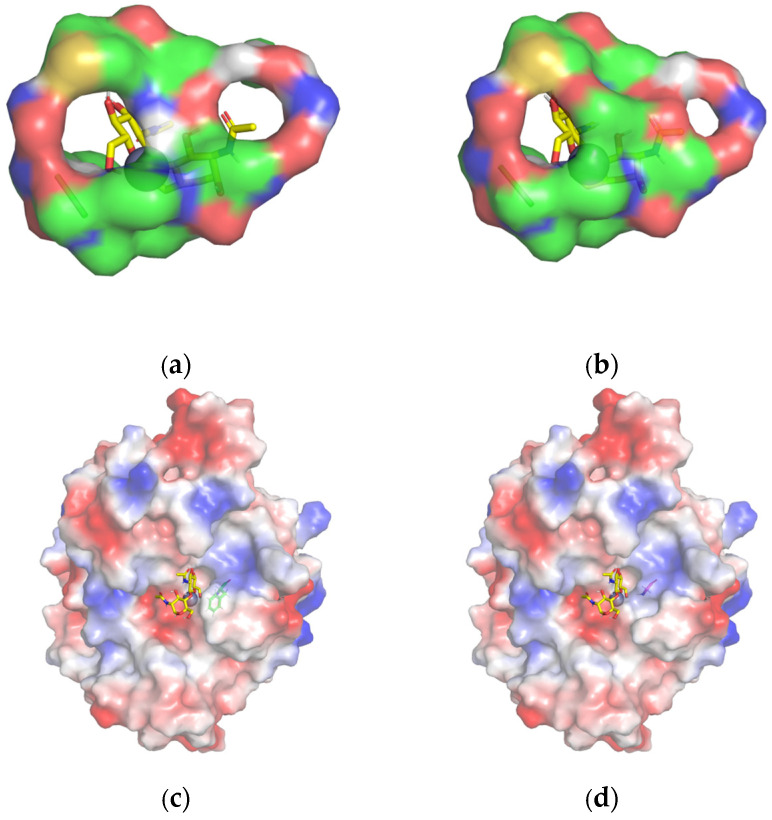
Protein ligand pocket of (**a**) wild-type and (**b**) F18G mutant; the spatial variation on the surface of the substrate-binding region of (**c**) wild-type and (**d**) F124G mutant. The balls indicate metal ions (grey) and (GlcNAc)_2_ (yellow) are represented as stick models. Red represents negative charges, and blue represents positive charges.

**Figure 8 ijms-25-02075-f008:**
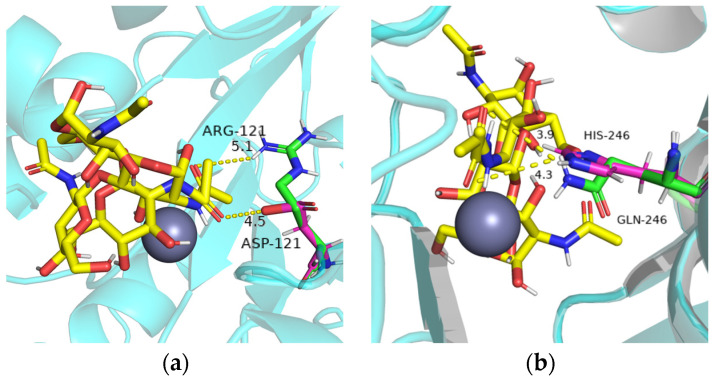
Mutant molecular docking diagram of *Es*CDA-9k. (**a**) R121D mutant; (**b**) Q246H mutant. The amino acid residues (green), mutant residues (pink) and (GlcNAc)_2_ (yellow) are represented as stick models. The grey balls indicate metal ions and cartoon mode shown in blue.

**Figure 9 ijms-25-02075-f009:**
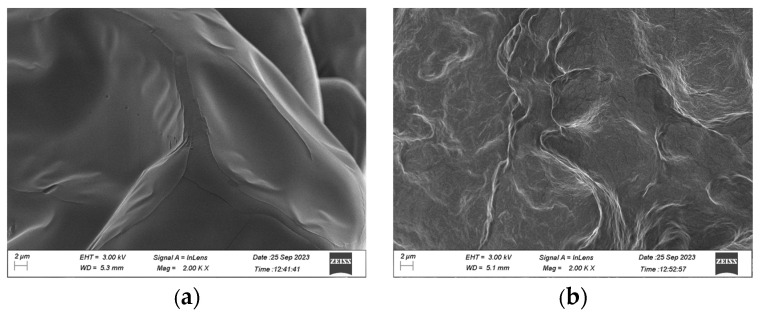
Scanning electron microscopy (SEM) images for (**a**) α-chitin and (**b**) *Es*CDA-9k-treated α-chitinwith magnificent × 2000 times.

**Figure 10 ijms-25-02075-f010:**
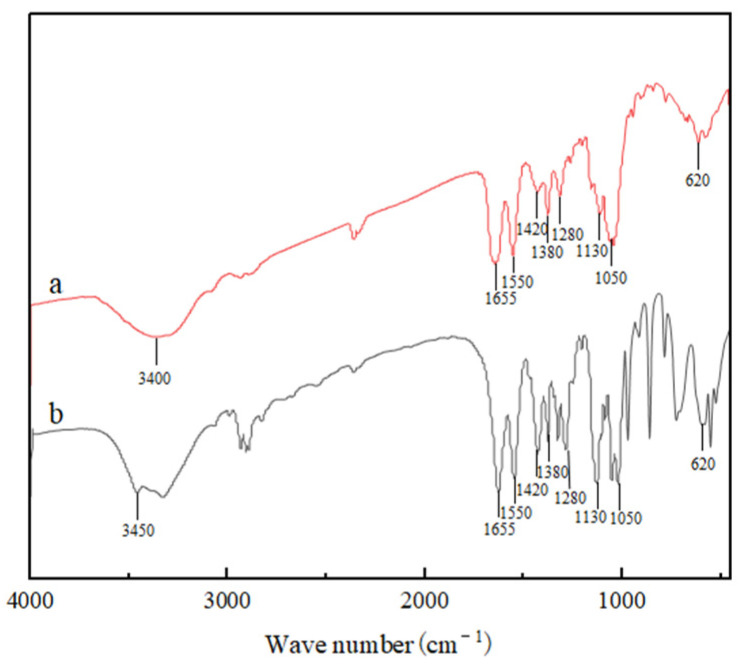
FT-IR spectra of (**a**) *Es*CDA-9k-treated α-chitin and (**b**) α-chitin.

**Table 1 ijms-25-02075-t001:** Purification of recombinant *Es*CDA-9k.

	Total Activity (U)	Protein Concentration (mg/mL)	Specific Activity (U/mg)	Purification Fold	Activity Yield (100%)
Crude enzyme	161.7	8.6	18.8	1	100
Ni^2+^-NTA	137.8	5.7	24.2	1.3	85.2

**Table 2 ijms-25-02075-t002:** Enzyme activity of wild-type and mutants.

	Enzyme Activity (U/mg)	Relative Enzyme Activity
Wild-type	24.2 ± 0.22	100%
F18G	38.7 ± 0.64	160%
R121D	30.1 ± 0.67	124%
F124G	34.2 ± 0.42	141%
Q246H	32.6 ± 0.86	135%

## Data Availability

Data is contained within the article and Appendix A.

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
