# Peer review of "Cloning and Characterization of Chitin Deacetylase from Euphausia superba"

_ijms, 2024, doi:10.3390/ijms25042075_

Round 1

Reviewer 1 Report

Comments and Suggestions for Authors

The article “Cloning and Characterization of Chitin Deacetylase from Euphausia superba” has high applied significance. This is the first report of the cloning and heterologous expression of the chitin deacetylase gene in Euphausia superba.

Authors identified and characterized a chitin deacetylase gene from Euphausia superba (EsCDA), recombinant enzyme was cloned, expressed, and purified. It was established that recombinant chitin deacetylase had the highest activity at 40 °C and pH 8.0 in Tris-HCl buffer. It was shown that EsCDA F18G mutant had a 1.6-fold higher activity than the wild-type of enzyme.

The article is written logically and well structured, but there are some minor comments:

1. Line 129: the authors write that chitin deacetylase exhibits good catalytic activity in Tris-HCl solution. From Figure 4c it is clearly seen that at pH 9.0 the enzyme activity in Tris-HCl buffer and glycine-NaOH buffer differs by approximately 3 times. The text of the article should provide an explanation of how the ionic composition of buffers can have such a significant effect on the activity of chitin deacetylase.

2. It is should to measure of chitin deacetylase activity at different substrate concentrations. And calculate parameters such as Km, Vmax, kcat.

3. It is worth explaining why the authors did not conduct experiments to determine the effects of temperature, pH, metal ions on the activity for mutants of chitin deacetylases, and also the effects of temperature and pH on the stability of mutant enzymes.

Technical shortcomings:

1. Lines 66-67: for some reason the authors provide a link to an article about cloning and characterization of fructose aldolase [36]. To reduce inappropriate self-citation, I propose to remove this sentence and this reference.

2. In the captions to Figures 4 and 5, it should be indicated under what conditions the enzyme activity taken as 100% was measured.

3. Line 281 and throughout the text: when listing its values multiple times, the unit of measurement should be written once, for example, instead of “20 °C, 25 °C, 30 °C, 35 °C, 40 °C, 45 °C, 50 °C, 55 °C" is better "20, 25, 30, 35, 40, 45, 50, 55 °C".

4. When describing IR spectroscopy, authors should indicate the type of device, resolution, number of cycles and scans.

Reviewer 2 Report

Comments and Suggestions for Authors

The work is devoted to the study of a new chitin deacetylase from Euphausia superba (EsCDA). The authors cloned the EsCDA gene, obtained a functional enzyme, and performed structural and functional analyzes using site-directed mutagenesis and in silico studies. The work is original, there is scientific novelty and practical application is justified. I am ready to recommend the manuscript for publication, but after major revision.

1.            The abbreviated name of the enzyme proposed by the authors has already been used to designate another enzyme (EsCDA-1 from Eriocheir sinensis, line 54). It needs to be renamed to avoid confusion.

2.            Line 77. 355 AA – Is it a sequence of mature protein or precursor? It must be explained.

3.            Lines 78-79. The charges of the residues are clearly mixed up. Asp and Glu are negatively charged, and Arg and Lys are positively charged.

4.            Line 86. “was found in Penaeus” need to change to “was found in CDAs from Penaeus”

5.            Lines 92-93. The picture uses several colors. Their use must be explained.

6.            Line 95. “optimized gene” – what do you mean? It must be explained.

7.            Lines 97-104: For what purpose do you use yeast instead of bacteria? Why don't you remove the tag (please specify what it is) from the target protein? Is the natural enzyme glycosylated?

To confirm the compliance of the resulting enzyme with the prototype, it is necessary to use some analytical methods. It is difficult to assess the purity of a protein by electrophoresis, only in a very rough approximation. You must confirm that the expression results in the product that you cloned. To confirm, you must do a mass spectrometric analysis.

8.            Table 1. If you used Ni2+-NTA, note it in Result and Methods

9.            Line 118. “temperature of Aspergillus “ need to change to “temperature of CDAs from Aspergillus “

10.          Line 135. “the optimum pH of Flammulina “need to change to “the optimum pH of CDAs from Flammulina “

11.          Lines 152-154. In this phrase, the phrase "In order to test" is used twice. Please rephrase.

12.          For what purpose are you assessing the effects of EDTA? Need to clarify. Is the set of ions arbitrary? It needs to be justified.

13.          Line 170. This is not a diagram

14.          Lines 172-173. The rationale for choosing the replacements is not clear. Some clarification is needed.

15.          Lines 180-188. This is not visible from the figure. Need modeling data to support your reasoning (how many and what kind of bonds, their number or length may be changed). In the current presentation, it turns out that modeling was done for the sake of pictures. This section needs to be rewritten.

16.          Lines 248-249. Do you mean the sequence you obtained? If yes, then you need to first write in detail how you obtained it. If not, the transcriptome sequencing project number for the organism must be provided.

17.          Lines 249-251. The proposal has not been formulated. How were the primers chosen, for which areas? Have you considered the signal peptide?

18.          Line 254. What primers were used to confirm the correctness of cloning? Figure S2b is misleading. All three bands need to be explained, not just the one you are looking for at 1600 bp.

19.          Lines 261-265. Provide a chromatogram, please.

20.          The Section of 3.7. needs reworking:

Line 315. “Based on the three-dimensional structure of AnCDA and CE4 deacetylases reported [52]”. What do you mean?

Lines 318-320. You should not write the number of microliters of reaction components in the mixture. It is much more important to provide a list of primers where substitutions are noted. Indicate which polymerases were used. Did you do the sequencing to validation?

Discussion section must be included.

Reviewer 3 Report

Comments and Suggestions for Authors This study involved the identification, cloning, and characterization of a chitin deacetylase gene (EsCDA) from Euphausia superba, resulting in a recombinant protein with a molecular weight of 45 kDa. The EsCDA, with a 1068 bp cDNA sequence encoding 355 amino acids, showed a 67.32% homology with Penaeus monodon. It exhibited optimal activity at 40°C and pH 8.0 in Tris-HCl buffer and responded variably to different metal ions. Notably, an EsCDA mutant (F18G) demonstrated 1.6 times higher activity than the wild type, marking this as the first report of EsCDA cloning and heterologous expression, providing a significant reference for future applications​   Overall the paper provides a comprehensive biochemical analysis of the enzyme Strengths Cloning and Bioinformatics Analysis: The paper successfully presents the cloning and characterization of the chitin deacetylase gene from Euphausia superba, providing a detailed bioinformatics analysis. This offers a significant contribution to the field by filling in the knowledge gap regarding chitin deacetylase in this specific species.   Optimization of Activity Conditions: The study meticulously investigates the effects of temperature and pH on the activity and stability of the enzyme EsCDA. This not only reveals the enzyme's optimal operating conditions but also contributes to understanding its stability under various environmental factors, which is crucial for its potential industrial applications.   Innovative Approach in Enzyme Activity Enhancement: The paper highlights the innovative use of site-directed mutagenesis to improve the enzyme's catalytic performance. By comparing the enzyme activity of mutants, it provides valuable insights into how specific amino acid changes can significantly enhance enzymatic activity, which is a substantial stride in enzyme engineering and biocatalysis research.   Weakness Lack of Comparative Analysis with Other Chitin Deacetylases: The paper provides a detailed study of the EsCDA enzyme from Euphausia superba but lacks a comparative analysis with chitin deacetylases from other organisms. Understanding the unique features or advantages of EsCDA in comparison to other chitin deacetylases is crucial for highlighting its potential applications and importance. The authors should consider including a section that compares the structural, kinetic, and functional properties of EsCDA with chitin deacetylases from other species. This could provide a broader context for the significance of their findings and potentially identify unique characteristics of EsCDA that could be exploited in industrial or biomedical applications.   Inadequate Investigation of the Enzyme's Industrial Applicability: While the paper successfully characterizes the EsCDA enzyme, it does not sufficiently explore its practical applicability, especially in industrial settings. For enzymes to be considered valuable for industrial processes, factors such as stability under various conditions, activity in the presence of different substrates or inhibitors, and scalability of production need to be thoroughly investigated. Future studies should aim to test the enzyme under conditions that mimic industrial processes, such as varying pH levels, presence of industrial solvents, or high substrate concentrations. Additionally, exploring the scalability of recombinant EsCDA production and its activity in industrial-relevant reactions would significantly enhance the paper's impact.   Limited Discussion on the Evolutionary Aspect of EsCDA:   The paper presents a bioinformatics analysis that includes homology modeling but does not delve into the evolutionary aspect of EsCDA, especially how its structure and function might have evolved in comparison to its homologs in other species. An evolutionary perspective could provide insights into the functional adaptations of the enzyme and its evolutionary trajectory. he authors should consider expanding the bioinformatics section to include an evolutionary analysis. This could involve constructing a phylogenetic tree to position EsCDA among chitin deacetylases from various species, followed by a discussion on the evolutionary pressures and adaptations that might have shaped its current structure and function.   Addressing these criticisms can provide a more comprehensive understanding of EsCDA, its potential applications, and its place in the broader context of chitin deacetylase research.   Comments on the Quality of English Language

Easy to read

Round 2

Reviewer 1 Report

Comments and Suggestions for Authors

Authors significantly corrected the text of the manuscript and responded to all my comments. I recommend publishing this article

Reviewer 2 Report

Comments and Suggestions for Authors

 Accept in present form